# NAT4AT: Using Non-Autoregressive Translation Makes Autoregressive Translation Faster and Better

## ABSTRACT

With the increasing number of web documents, the demand for translation has increased dramatically. Non-autoregressive translation (NAT) models can significantly reduce decoding latency to meet the growing translation needs, but they sacrifice translation quality. And there is still an irreparable performance gap between NAT models and strong autoregressive translation (AT) models at the corpus level. However, more fine-grained comparative experiments on AT and NAT are currently lacking. Therefore, in this paper, we first conducted analysis experiments at the sentence level and found complementarity and high similarity between the translations generated by AT and NAT. Then, based on this observation, we propose a general and effective method called NAT4AT, which can not only use NAT to speed up the inference speed of AT significantly but also improve its final translation quality. Specifically, NAT4AT first uses a NAT model to generate an original translation in parallel and then uses an AT model as a correction model to revise errors in the original translation. In this way, the AT model no longer needs to predict the entire translation but only needs to predict a small number of error parts in the NAT result. Extensive experimental results on major WMT benchmarks verify the generality and effectiveness of our method, whose translation quality is superior to the strong AT model and achieves a **5.0×** speedup.

## CCS CONCEPTS

• **Computing methodologies** → **Machine translation**; **Natural language generation**.

## KEYWORDS

Neural Machine Translation, Non-Autoregressive Generation, Efficient Inference

**ACM Reference Format:**
Anonymous Author(s). 2023. NAT4AT: Using Non-Autoregressive Translation Makes Autoregressive Translation Faster and Better. In *Proceedings of Make sure to enter the correct conference title from your rights confirmation emai (Conference acronym 'XX).* ACM, New York, NY, USA, 12 pages. https://doi.org/XXXXXXX.XXXXXXX

## 1 INTRODUCTION

Web data is multilingual, and its volume is rapidly increasing. To better mine and utilize the web data, neural machine translation

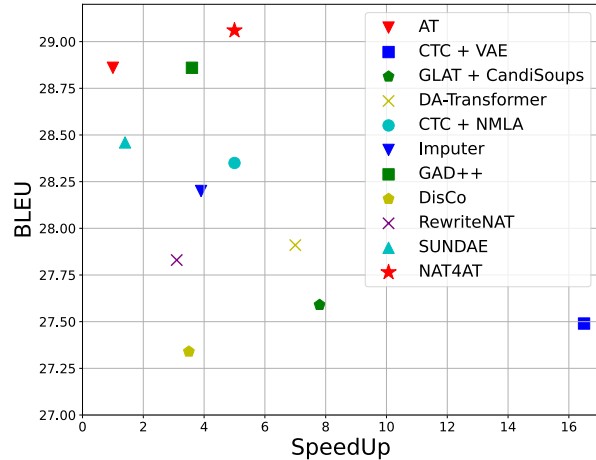

**Figure 1: Efficiency (SpeedUp) and Translation quality (BLEU) of AT and NAT models in the WMT'14 EN→DE translation task.**

(NMT) models are essential, which require strong translation capabilities and fast inference speeds. Transformer-based models [34] have been widely used in NMT, and they usually adopt an autoregressive decoding paradigm where each generation step depends on previously generated tokens. The autoregressive translation (AT) model can be stably trained using the teacher forcing method and produce high-quality translations. However, it strictly models left-to-right dependencies between target tokens, resulting in poor inference efficiency, especially for long sentences. To alleviate this problem, non-autoregressive translation (NAT) is proposed [6], which assumes that the target tokens are conditionally independent and can be decoded in parallel, significantly improving inference efficiency. However, NAT suffers from performance degradation due to the multi-modality problems [6].

Significant efforts [5, 11, 13, 20, 36] have been made to improve NAT performance and narrow the gap between NAT and AT models. Although these methods can effectively improve translation quality, they still cannot fully solve the inherent multi-modality problem of NAT, which results in an irreparable gap between NAT and strong AT models.

While existing work shows that NAT performs worse than AT, this is a corpus-level comparison. From a more fine-grained perspective, we still have the following two questions:

(1) How similar are the translations generated by AT and NAT for the same source sentence?

(2) Do AT and NAT have their own strengths for different source sentences?

To the best of our knowledge, no previous work has explored these two questions in detail. Therefore, we conduct a preliminary

experiment in Section 3 to compare AT and NAT models at the sentence level. Experimental results demonstrate a high overlap rate between the translations generated by NAT and AT models. In addition, for some sentences, NAT models can translate better than AT, suggesting that the translation capabilities of NAT and AT models can be complementary. Therefore, based on this observation, we propose a simple and effective method called NAT4AT, which can not only significantly accelerate the inference speed of AT by using NAT but also improve the final translation quality.

Similar to our method, recently, Xia et al. proposed generalized aggressive decoding (GAD), which can speed up AT without quality loss through NAT. However, GAD needs to specifically design and train a NAT drafter model, and strictly requires that the final translation is exactly the same as the AT. When there is a bifurcation, GAD discards all subsequent tokens generated by the NAT drafter and regenerates them in the next iteration. However, due to NAT's conditional independence assumption, an incorrect prediction at the current position does not necessarily imply that subsequent predicted tokens will also be incorrect. Unlike GAD, NAT4AT only requires NAT model decoding once to generate the original translation and then uses the AT model as a correction model to revise errors in the original translation. Furthermore, NAT4AT does not require the final translation to be identical to AT. Therefore, compared to GAD, our method has three advantages: 1) it is orthogonal to the model, meaning that any NAT model can be used to generate the original translation; 2) the NAT model only needs to decode once, making the inference process more efficient; 3) by leveraging the complementarity of NAT and AT models, it can achieve better performance than AT.

Extensive experimental results on major WMT benchmarks demonstrate the generality and effectiveness of our method. For various NAT and AT models, our method can not only generate higher-quality translations than strong AT models, but also significantly improve the inference efficiency. More notably, as shown in Figure 1, our best variant achieves **29.06** BLEU with **5.0×** speedup in the WMT'14 EN→DE translation task.

## 2 BACKGROUND

### 2.1 Autoregressive Translation

The autoregressive translation (AT) model achieves state-of-the-art performance on multiple machine translation benchmarks. Given a source sentence $X = (x_1, x_2, ..., x_n)$ and a target sentence $Y = (y_1, y_2, ..., y_m)$, the AT model decomposes the target distribution of translation into a chain of conditional probabilities with a left-to-right causal structure:

$$P_{\text{AT}}(Y \mid X) = \prod_{i=1}^{m} P\left(y_i \mid y_{<i}, X; \theta_{\text{AT}}\right)$$

where $y_{<i}$ denotes the previous target tokens before the $i^{th}$ position. During training, the AT model can perform parallel decoding through the teacher-forcing strategy. However, during inference, the AT model cannot obtain target tokens, so it needs to generate tokens one by one from left to right until **[EOS]** is generated.

Therefore, while AT models can achieve desirable translation quality, their sequential decoding strategy during inference does not

**Table 1: The corpus-level BLEU score of AT and NAT models in the WMT'14 EN→DE and WMT'14 DE→EN test sets. Max(AT, NAT) indicates that the final translation of each sentence is the one with a higher BLEU among the results generated by AT and NAT.**

| Model | WMT'14 | |
|---|---|---|
| | EN→DE | DE→EN |
| AT (Transformer-Base) | 27.92 | 31.40 |
| GLAT | 25.33 | 29.04 |
| GLAT + CandiSoups | 27.67 | 31.04 |
| Max(AT, GLAT) | 29.61 | 33.63 |
| Max(AT, GLAT + CandiSoups) | 30.28 | 34.16 |

fully use existing parallel computing hardware, which is inefficient when translating long sentences.

### 2.2 Non-Autoregressive Translation

To reduce decoding latency, the non-autoregressive translation (NAT) model is proposed [6]. The Vanilla NAT model abandons explicit modeling of left-to-right dependencies between target tokens, instead assuming that target tokens only depend on the source sentence $X$, so all tokens can be generated in parallel:

$$P_{\text{NAT}}(Y \mid X) = \prod_{i=1}^{m} P\left(y_i \mid X; \theta_{\text{NAT}}\right)$$

This generation paradigm can fully use existing hardware's computing power to improve inference efficiency significantly. However, due to the lack of dependency information between target tokens, NAT has a serious multi-modality problem [6], resulting in a significant decrease in translation quality. Although many advanced NAT methods [13, 20, 36] have been proposed, there is still an irreparable performance gap between NAT and strong AT models due to the inherent flaws of NAT itself.

## 3 PRELIMINARY EXPERIMENT

This section explores AT and NAT models at the more fine-grained level. Specifically, we conduct experiments to answer two main questions: 1) For the same source sentence, how similar are the translations generated by AT and NAT? 2) For different source sentences, do AT and NAT have their strengths?

### 3.1 Experimental Setup

We conduct experiments on two translation tasks, WMT14 EN→DE and WMT14 DE→EN. For AT, we use a conventional autoregressive Transformer. For NAT, we choose GLAT [20] and GLAT+CandiSoups [36], two methods with different performances. Both AT and NAT models are based on the Transformer-Base architecture and trained using open-source distilled data [12]. During inference, the beam size is set to 5 for AT, and the number of candidate translations is set to 5 for the CandiSoups algorithm. The performance of AT and NAT models is shown in Table 1. In the experiment, we use sentence-level BLEU to judge the translation quality of different models for different source sentences. We use the overlap rate to

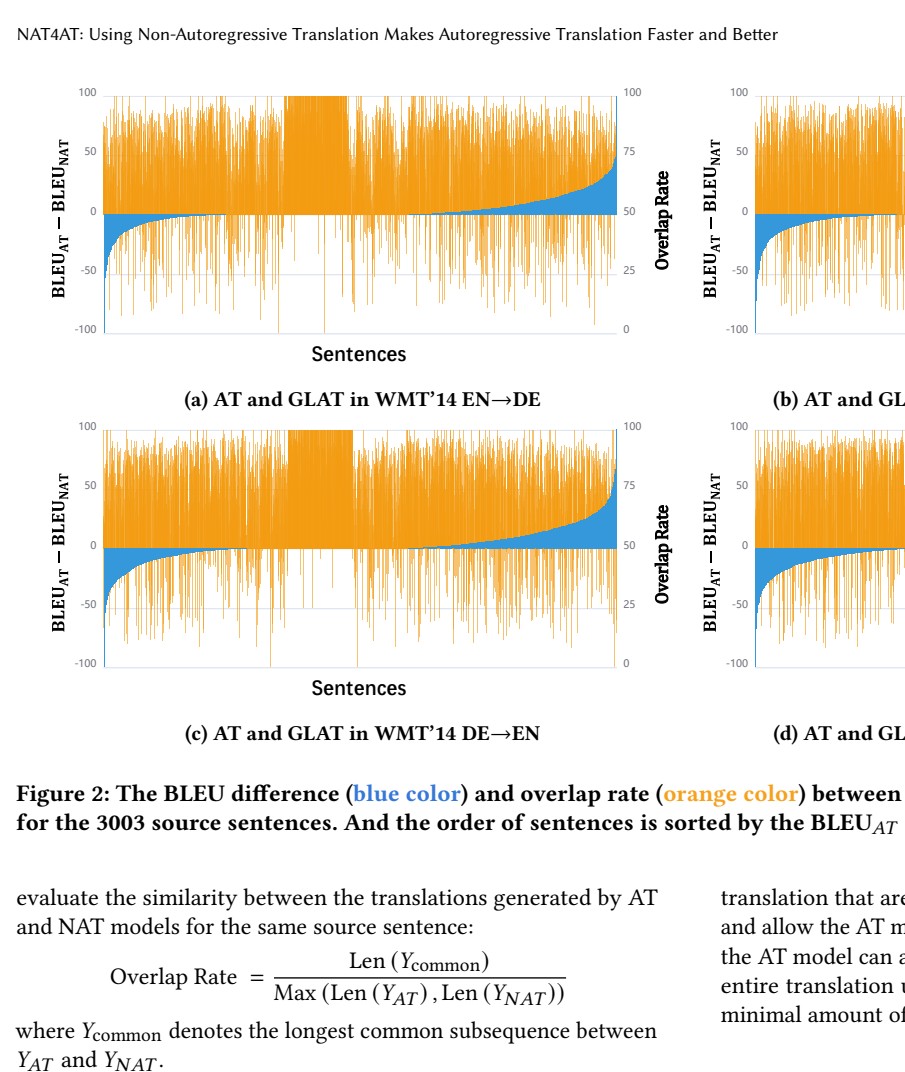

(a) AT and GLAT in WMT'14 EN→DE

(b) AT and GLAT+CandiSoups in WMT'14 EN→DE

(c) AT and GLAT in WMT'14 DE→EN

(d) AT and GLAT+CandiSoups in WMT'14 DE→EN

**Figure 2: The BLEU difference (blue color) and overlap rate (orange color) between the translations generated by AT and NAT for the 3003 source sentences. And the order of sentences is sorted by the $BLEU_{AT}$ - $BLEU_{NAT}$.**

evaluate the similarity between the translations generated by AT and NAT models for the same source sentence:

$$\text{Overlap Rate } = \frac{\text{Len}\left(Y_{\text{common}}\right)}{\text{Max}\left(\text{Len}\left(Y_{AT}\right), \text{Len}\left(Y_{NAT}\right)\right)}$$

where $Y_{\text{common}}$ denotes the longest common subsequence between $Y_{AT}$ and $Y_{NAT}$.

### 3.2 Experimental Results

The main experimental results are shown in Figure 2. As we can see, the translations generated by the AT model and the two NAT models are complementary and highly similar. Specifically, for the same source sentence, the average overlap rate of translations generated by AT and NAT is above 70%, and the stronger NAT model can achieve higher overlap rates. For example, the average translation overlap rate between GLAT+CandiSoups and AT on the WMT14 DE→EN task reached 77.22%. In addition, the performance of AT and NAT models is also different for different source sentences. And the stronger the NAT model, the more source sentences it can translate better than the AT model. In the WMT'14 EN→DE translation task, GLAT+CandiSoups achieves higher translation quality than AT on 38.79% source sentences. What's more, Table 1 shows that if the better translation generated by AT and NAT models is selected for each sentence, the corpus-level BLEU can be significantly improved. Even using GLAT, which has a large performance gap with the AT model, can still achieve 1.69 and 2.23 BLEU improvements on the AT model.

The above experimental results demonstrate that the NAT model has the potential to make the AT model faster and better. The key to achieving this goal is to retain the parts of the NAT generated

translation that are better than or the same as the AT translation and allow the AT model to revise only the wrong parts. In this way, the AT model can achieve better performance than generating the entire translation using itself independently and requires only a minimal amount of iterative decoding.

## 4 APPROACH

This section details the method proposed in this paper. We first present the problem definition in 4.1, then introduce the implementation details of NAT4AT in Section 4.2, followed by a concrete example in Section 4.3.

### 4.1 Problem Definition

Given any trained NAT model, it can predict the original translation $Y^1 = (y_1^1, y_2^1, ..., y_m^1)$ in parallel. Given any trained AT model, it can autoregressively predict translation $Y^* = (y_1^*, y_2^*, ..., y_m^*)$ from left to right. We aim to use the AT model to verify and revise the original translation and generate a better result. Specifically, when $Y^1$ is input to the AT model, the AT model needs to judge from left to right whether $y_i^1$ is correct. If it is correct, the token will be retained. Otherwise, the AT model needs to revise this token with its own prediction. Moreover, since error propagation will have a huge impact on the subsequent prediction of the AT model, after the revision, it is necessary to construct the next iteration input $Y^{t+1}$ according to the current result and the original translation. Therefore, there are two key issues to be addressed: 1) How to determine if $y_i^t$ is correct; 2) How to generate $Y^{t+1}$ that needs to be verified in the next iteration.

| | |
|---|---|
| **Source Input** | Von den neuen Einschränkungen sind junge Menschen , Minderheiten und Menschen mit niedrigem Einkommen unverhältnismäßig stark betroffen . |
| **NAT Result $Y^1$** | The new restrictions [t-1 Window] disproportionately affect people , minorities and [t-3 Window] those those with low incomes . |

| | | |
|---|---|---|
| **AT Input $Y^1$** | The new restrictions disproportionately affect people , minorities and those those with low incomes . | |
| **AT Result $Y^*$** | The new restrictions inappropriately affect young , minorities and person with with low incomes . | t = 1 |
| $Y^2$ | The new restrictions disproportionately affect young , minorities and those those with low incomes . | |
| **AT Input $Y^2$** | The new restrictions disproportionately affect young , minorities and those those with low incomes . | |
| **AT Result $Y^*$** | The new restrictions disproportionately affect young people minorities and those person with low incomes . | t = 2 |
| $Y^3$ | The new restrictions disproportionately affect young people , minorities and those those with low incomes . | |
| **AT Input $Y^3$** | The new restrictions disproportionately affect young people , minorities and those those with low incomes . | |
| **AT Result $Y^*$** | The new restrictions disproportionately affect young people , minorities and those with with low incomes . | t = 3 |
| $Y^4$ | The new restrictions disproportionately affect young people , minorities and those with low incomes . | |
| **AT Input $Y^4$** | The new restrictions disproportionately affect young people , minorities and those with low incomes . | |
| **AT Result $Y^*$** | The new restrictions disproportionately affect young people , minorities and those with low incomes . | t = 4 |
| **Final Result** | The new restrictions disproportionately affect young people , minorities and those with low incomes . | |

**Figure 3: An example illustrates how NAT4AT generates high-quality final translations using both NAT and AT models. For this example, the s is set to 0.3, and w is set to 2. The highlighted token represents the token added to the final translation, and the blue token represents the token generated by NAT is the same as AT. For brevity, we omit [BOS] and [EOS] in this figure.**

---

**Algorithm 1: NAT4AT**

**Input:** Source sentence $\mathbf{X}$, $\theta_{NAT}$, $\theta_{AT}$, $\mathbf{s}$, $\mathbf{w}$
$Y^1 = (y_1^1, ..., y_m^1) = \arg\max P_{NAT}(Y|X; \theta_{NAT})$
$Y^* = (y_1^*, ..., y_m^*) = \arg\max P_{AT}(Y|X; Y^1; \theta_{AT})$
Initialize $i \leftarrow 1$, $t \leftarrow 1$
**while** $y_i^* \neq$ *[EOS]* **and** $i <$ *MAX_LEN* **do**
  **if** $P_{AT}(y_i^*) - P_{AT}(y_i^t) < s$ **then**
    | $i \leftarrow i + 1$
  **else**
    **if** $\exists j \in [i - w, i + w]$ **and** $y_j^1 = y_i^*$ **then**
      | $Y^{t+1} \leftarrow$ CAT$(y_{<i}^t, y_i^*, y_{>j}^1)$
    **else**
      | $Y^{t+1} \leftarrow$ CAT$(y_{<i}^t, y_i^*, y_{>i}^t)$
    **end**
    $Y^* = \arg\max P_{AT}(Y|X; Y^{t+1}; \theta_{AT})$
    $t \leftarrow t + 1$
  **end**
**end**
return $Y^t[:i]$

---

## 4.2 NAT4AT

For the above problems, we propose a simple but very effective algorithm called NAT4AT, which is summarized in Algorithm 1. It greatly preserves the original translation predicted by NAT and dramatically reduces the number of iterative decoding, thus significantly accelerating the inference speed.

First, NAT4AT will input $Y^t$ to the AT decoder at the t-th iteration to obtain the probability distribution output by the AT model and the corresponding translation:

$$P_{AT}(Y \mid X; Y^t) = \prod_{i=1}^{m} P\left(y_i \mid X; y_{<i}^t; \theta_{AT}\right)$$

$$Y^* = (y_1^*, ..., y_m^*) = \arg\max P_{AT}(Y \mid X; Y^t)$$

Note that since $Y^t$ is known, the AT model can decode in parallel like the training phase. Then NAT4AT will compare $P_{AT}(y_i^t \mid X; y_{<i}^t)$ and $P_{AT}(y_i^* \mid X; y_{<i}^t)$ from left to right. If the difference between the two is less than the threshold $\mathbf{s}$, NAT4AT will consider the current NAT prediction result is reliable and keep $y_i^t$. Otherwise, NAT4AT will use $y_i^*$ to revise $y_i^t$. In this way, NAT4AT does not need to strictly guarantee that the final translation result is the same as $Y^*$, thus retaining more original translations, enabling NAT to supplement AT's translation performance, and reducing the number of AT decoder iterations.

For the second problem, due to NAT's conditional independence assumption, the token at the current position is incorrectly predicted, which does not mean that subsequent tokens are also wrong. Moreover, there is likely to be a misalignment between AT and NAT generated translations, i.e., even if $y_i^*$ is different from $y_i^1$, it may be the same as other tokens nearby (as shown in Figure 3). Therefore, we propose a sliding window algorithm to find the corresponding position of $y_i^*$ in $Y^1$. Specifically, if the window width is set to $\mathbf{w}$, NAT4AT will determine whether exists $j$ in $[i - w, i + w]$ that satisfies $y_j^1 = y_i^*$. If it exists, NAT4AT will concatenate $y_{>j}^1$ after $y_i^*$ to generate $Y^{t+1}$, otherwise NAT4AT will concatenate $y_{>i}^t$ after $y_i^*$ to form $Y^{t+1}$. Finally, when AT verifies to the [EOS] token, iteration terminates. Our experimental results show that this sliding window algorithm can make maximum use of the original translation, significantly reducing the number of iterations.

## 4.3 Example

Figure 3 shows an example of how NAT4AT generates high-quality translations with the cooperation of AT and NAT models. First, when t=1, NAT4AT takes the original translation $Y^1$ generated by the NAT model as input to the AT decoder and obtains the AT result. Then, by traversing from left to right, NAT4AT finds that

**Table 2: Comparison between our method and existing methods on WMT'14 EN↔DE and WMT'16 EN↔RO benchmarks. The AT baseline's results are cited from Qian et al. 20, and all NAT baselines' results reported are quoted from respective papers. Iter. is the average number of decoding iterations, Adv. denotes adaptive determines the number of iterations, * means models trained with distillation data from Transformer-Big. And the † denote NAT4AT is statistically better (t-test with p-value < 0.01) than strong Transformer-Base (beam=5).**

| Models | | Iter. | Speedup | WMT'14 | | WMT'16 | |
|---|---|---|---|---|---|---|---|
| | | | | EN→DE | DE→EN | EN→RO | RO→EN |
| AT baseline | Transformer-Base (w/o Seq-KD) | N | 1.0× | **27.48** | **31.27** | **33.70** | **34.05** |
| Iterative NAT | InsT [33] | ≈log N | 4.8× | 27.41 | - | - | - |
| | CMLM [5]* | 10 | 1.7× | 27.03 | 30.53 | 33.08 | 33.31 |
| | DisCO [15]* | Adv. | 3.5× | 27.34 | 31.31 | 33.22 | 33.25 |
| | Imputer [25]* | 8 | 3.9× | 28.20 | 31.80 | 34.40 | 34.10 |
| | SUNDAE [26] | 16 | 1.4× | 28.46 | 32.30 | - | - |
| | RewriteNAT [4]* | 2.7 | 3.1× | 27.83 | 31.52 | 33.63 | 34.09 |
| | CMLMC [14]* | 10 | - | 28.37 | 31.41 | 34.57 | 34.13 |
| | GAD [35]* | 4.9 | 3.0× | 28.73 | 32.18 | 34.83 | 34.65 |
| | GAD++ [35]* | 4.0 | 3.6× | **28.89** | **32.56** | **35.32** | **34.98** |
| Fully NAT | Vanilla NAT [6] | 1 | 15.6× | 17.69 | 21.47 | 27.29 | 29.06 |
| | ReorderNAT [22] | 1 | 16.1× | 22.79 | 27.28 | 29.30 | 29.50 |
| | OAXE [3]* | 1 | 15.3× | 26.10 | 30.20 | 32.40 | 33.30 |
| | CTC + VAE [7] | 1 | 16.5× | 27.49 | 31.10 | 33.79 | 33.87 |
| | GLAT [20] | 1 | 15.3× | 25.21 | 29.84 | 31.19 | 32.04 |
| | GLAT + Candidate Soups [36] | 1 | 7.8× | 27.59 | 30.95 | 33.22 | 33.73 |
| | DA-Transformer [13]* | 1 | 7.0× | 27.91 | 31.95 | - | - |
| | CTC + NMLA [28] | 1 | 5.0× | **28.35** | **32.27** | **34.72** | **34.95** |
| Ours | AT Teacher (beam = 5) | N | - | 29.32 | 32.66 | 34.82 | 34.89 |
| | Transformer-Base (beam = 5)* | N | 1.0× | 28.86 | 32.26 | 35.16 | 35.33 |
| | Transformer-Base (beam = 1)* | N | 1.1× | 28.69 | 32.20 | 35.01 | 35.18 |
| | NAT Model (GLAT + Candidate Soups)* | 1 | 7.8× | 28.02 | 31.49 | 34.30 | 34.58 |
| | NAT4AT | 4.3 | 5.0× | 29.06† | 32.58† | 35.39† | 35.66† |

**Table 3: COMET-22 scores on WMT'14 DE→EN and WMT'16 RO→EN test sets.**

| Model | DE→EN | RO→EN |
|---|---|---|
| Transformer-Base (beam=1) | 79.49 | 63.92 |
| Transformer-Base (beam=5) | 79.56 | 64.01 |
| GLAT + CandiSoups | 77.63 | 63.62 |
| NAT4AT | **79.73** | **64.23** |

the predicted probability difference of the first five tokens is below the threshold of 0.3, so they are kept. And when i=6, $P_{AT}$("young")-$P_{AT}$("people") is greater than 0.3, so NAT4AT believes that "people" is incorrectly predicted by NAT, and replaces it with "young." Then, because "young" does not exist in the sliding window, NAT4AT concatenates $y_{>6}^t$ after "young" to form $Y^2$.

When t=2, since the AT model has causal masks, the prediction results in the first six positions are the same as before, so they can be skipped directly. And when i=7, $P_{AT}$("people")-$P_{AT}$(",") is greater than 0.3, so NAT4AT revises "," with "people". Then NAT4AT finds "people" exist in the sliding window, so it concatenates $y_{>6}^1$ after

"people" to form $Y^3$. Similarly, when t = 3, NAT4AT revises the first error in $Y^3$ and generates $Y^4$ based on $Y^1$. When t=4, NAT4AT traverses to the end and finds no errors in $Y^4$, so the iteration terminates, and $Y^4$ is the final result. From this example, we can find that NAT4AT can not only retain most of the correct tokens in the original translation but also solve the problems of mistranslation, missing translation, and repeated translation through a small amount of AT iterative decoding.

## 5 EXPERIMENTS

In this section, we first introduce the experimental setup in Section 5.1, then report the main results in Section 5.2. Analysis experiments are presented in Section 5.3.

### 5.1 Experimental Setup

*Datasets and Evaluation.* We validate our proposed method on two standard translation benchmarks that are widely used in previous studies, i.e., WMT'14 English (EN)↔German (DE) (4.0M pairs) and WMT'16 English (EN)↔Romanian (RO) (0.6M pairs). And for EN↔DE and EN↔RO, we apply the same prepossessing steps and

learn joint BPE subwords [27] vocabulary as mentioned in previous work (EN↔DE: Zhou et al., EN↔RO: Xia et al.). For a fair comparison, we use tokenized BLEU [19] for all benchmarks to evaluate translation quality. In addition, we use COMET-22 [24], a learned and reference-based metric, as a complement to evaluate translation quality. To evaluate inference efficiency, the speedup and the average number of decoding iterations are measured with a batch size of 1 on the WMT'14 En→De test set. Our experiments are conducted on Nvidia A100-40G GPUs, and we use the Fairseq [18] to implement our method.

*Knowledge Distillation.* Following the previous work [34, 35, 37], we employ sequence-level knowledge distillation for our AT and NAT models in all datasets. Specifically, for WMT'14 EN↔DE, we use Transformer-Big as the teacher and generate distillation data with beam search (beam = 5). And for WMT'16 EN↔RO, we use Transformer-Base as the teacher and also set beam = 5.

*Implementation Details.* We use the conventional autoregressive Transformer as our AT correction model, and GLAT+CandiSoups [36] as our NAT model to generate the original translation. Moreover, since the CandiSoups algorithm requires using the AT model to re-score NAT generated translations, we directly use the probability distribution generated by CandiSoups in the first iteration of NAT4AT. All our models use the hyperparameters of Transformer-Base [34]. During training, all AT models are trained for 100k updates with a batch of 256k tokens, and all NAT models are trained for 200k updates with a batch of 128k tokens. We adopt the Adam optimizer [16] with $\beta = (0.9, 0.98)$. And the learning rate warms up to $5 \cdot 10^{-4}$ within 4k steps and then decays with the inverse square schedule. For regularization, we use dropout (WMT'14 EN↔DE: 0.1, WMT'16 EN↔RO: 0.3), 0.001 weight decay, and 0.1 label smoothing. We evaluate the BLEU scores on the validation set every 500 steps and average the best 5 checkpoints for the final model. During inference, we use a beam size of 5 for the AT baseline, use 5 candidate translations for the CandiSoups algorithm, and set threshold **s** = 0.35 and window width **w** = 8 for our NAT4AT method.

## 5.2 Main Results

As shown in Table 2, our proposed NAT4AT method can achieve superior performance in the trade-off between translation quality and inference speed. We highlight the advantages of our method compared to other methods:

First, our method can generate better translations than all previous Fully NAT methods. Even compared to CTC+NMLA [28], NAT4AT outperforms it on all translation tasks and maintains the same inference speed, verifying that NAT4AT can effectively use the AT model to revise the wrong parts in NAT translation, thus achieving the performance that NAT is difficult to achieve.

Second, our method can achieve lower inference latency than previous Iterative NAT methods. GAD++ [35] can achieve superior performance by combining AT and NAT models. However, NAT4AT can outperform GAD++ regarding translation quality and inference speed. Note that although NAT4AT requires an average of 0.3 more iterations than GAD++, NAT4AT only needs to use the AT decoder during iterations, while GAD++ requires both NAT and AT decoders, so NAT4AT's inference latency is lower than

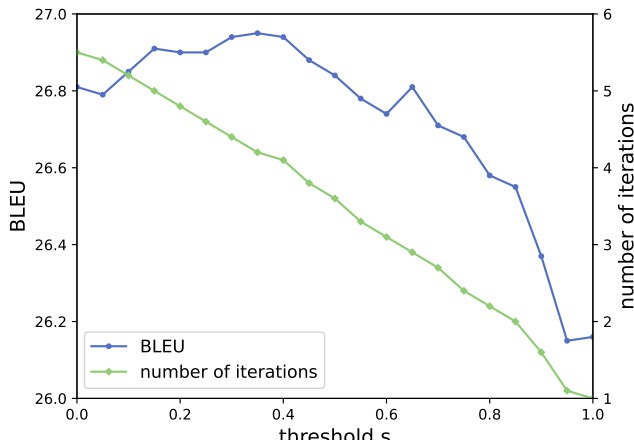

**Figure 4: Performance of NAT4AT with different threshold s on WMT'14 EN→DE validation set.**

GAD++. Moreover, because NAT4AT takes advantage of the complementarity between AT and NAT translations, it can achieve better translation quality than GAD++.

Finally, our method can make AT models translate faster and better. Specifically, NAT4AT achieves higher BLEU scores than strong AT models with Seq-KD on all four translation tasks while maintaining a 5.0× speedup. The significance test results also prove that using NAT4AT can statistically achieve better results than the strong AT models. Considering that BLEU may be biased, we also measure our method with COMET-22 on WMT'14 DE→EN and WMT'16 RO→EN tasks. As shown in Table 3, NAT4AT's performance is still better than the strong AT model.

In conclusion, the above experimental results demonstrate that our method can fully exploit the similarity and complementarity between AT and NAT translations, thus achieving excellent translation quality and inference speed. Moreover, NAT4AT is orthogonal to the model, so it can achieve better performance while using more advanced AT and NAT models.

## 5.3 Analysis

### 5.3.1 Hyperparameter.

*Influence of the threshold s.* We set the window width **w** = 8 unchanged and conduct experiments with various threshold **s** on WMT'14 EN→DE validation set and show the results in Figure 4. As we can see, **s** significantly affects translation quality and the number of iterations. Specifically, when **s** = 0, NAT4AT requires the translation generated by the NAT model to be exactly the same as AT, so more iterations are needed. As **s** increases, the number of iterations required by NAT4AT decreases linearly. And because of the complementarity between NAT and AT translations, the translation quality is improved. However, when **s** > 0.4, NAT4AT may retain some wrong tokens generated by the NAT model, resulting in a gradual deterioration of its performance. Finally, when **s** = 1, NAT4AT no longer uses AT for correction, so its performance is exactly the same as the NAT model.

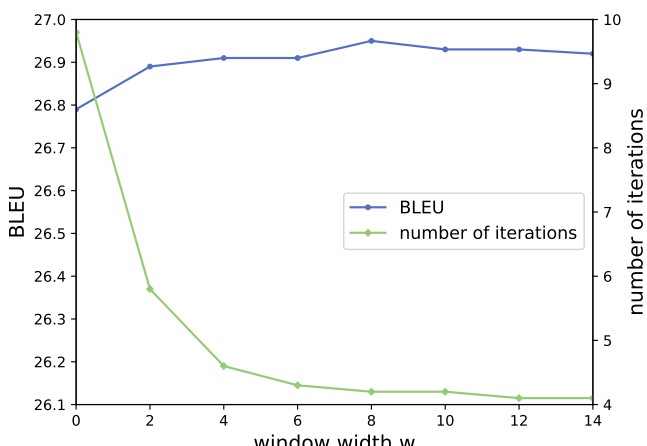

**Figure 5: Performance of NAT4AT with different window width w on WMT'14 EN→DE validation set.**

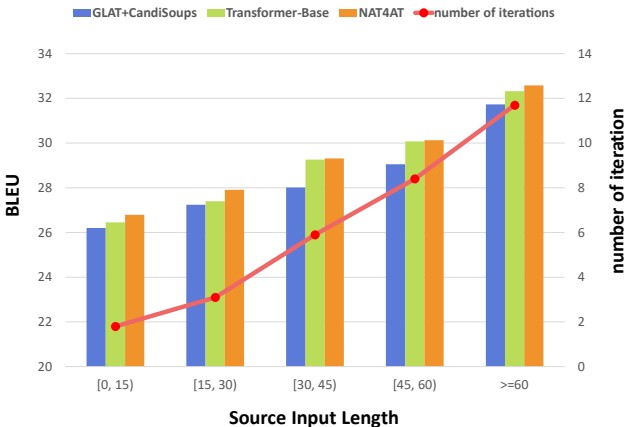

**Figure 6: Performance under different source sentence lengths on WMT'14 EN→DE test set.**

This experimental result shows that we can easily trade off translation quality and inference speed by choosing different **s** according to different scenarios. Low **s** can achieve high translation quality, and high **s** can achieve fast inference speed. And we set the threshold **s** to 0.35 in our experiments.

*Influence of the window width w.* To analyze the effect of window width **w** on our method, we conduct experiments with different **w** on WMT'14 EN→DE validation set. Figure 5 shows that our proposed sliding window algorithm can significantly reduce the number of iterations and improve translation quality. When **w** = 0, NAT4AT strictly restricts the translation of AT and NAT to be completely aligned, which does not fully use the original translation generated by NAT, so an average of 9.8 iterations are required during the inference process. However, when **w** = 2, NAT4AT allows misalignment between AT and NAT translations, significantly reducing the number of iterations (9.8 → 5.8). In addition, the translation quality is improved because more NAT results are used.

When **w** > 8, the number of iterations does not obviously decrease with the increase of **w**. And due to the expansion of the sliding window, there may be multiple identical tokens in the window, resulting in alignment errors between AT and NAT, so the translation quality of our method drops slightly. Therefore, in our experiments, we set the window width **w** to 8.

*5.3.2 NAT and AT Models with Different Performance.* Our proposed NAT4AT is a general approach that can be applied to arbitrary NAT and AT models. Therefore, we also conducted experiments based on different NAT and AT models to verify the generality and effectiveness of our method. As shown in Table 4, NAT4AT can perform excellently in all scenarios. Specifically, when using GLAT, which has a large performance gap compared to AT, to generate the original translation, NAT4AT can still achieve better performance than AT. However, because there are more errors in the original translation, NAT4AT requires more iterations. When using a stronger NAT model (GLAT-Big w/ CandiSoups) and AT model (Transformer-Big), NAT4AT can achieve superior performance, with BLEU scores of **29.43** and **32.91** on WMT'14 EN↔DE.

These experimental results show that the performance of NAT and AT models affects NAT4AT's translation quality and inference speed. When using stronger NAT and AT, NAT4AT can generate better translations. And the smaller the performance difference between NAT and AT models, the faster the NAT4AT inference speed. This further confirms the potential of our proposed method, which can benefit from more advanced AT and NAT methods to achieve better performance in the future.

*5.3.3 Influence of the Source Input Length.* We also explore the effect of sentence length on the performance of our method. Specifically, we partition the source sentences of the WMT'14 EN→DE test set into five intervals by the length after BPE and compute the BLEU score and iteration number for each interval. The experimental results are shown in Figure 6. As we can see, when the source sentence length increases, the translation quality of both two models increases, and the AT model consistently outperforms the NAT model. These results suggest that the sentences that AT and NAT models are respectively good at translating cannot be distinguished directly by length. And we believe that it is an interesting research direction to explore the types of sentences that AT and NAT excel at translating, and we will explore this question in the future. However, our method can still outperform the AT model at each interval, which shows that NAT4AT can effectively distinguish the quality of NAT generated tokens and exploit its complementarity with AT translation to generate better translations.

Moreover, although the average number of iterations required by NAT4AT increases as the length of the source sentence increases, it is only 10%-20% of the iterations necessary by the AT model. This result shows that NAT4AT can take advantage of the high similarity between AT and NAT translations to significantly reduce the number of iterations and improve inference efficiency.

*5.3.4 Influence of the Knowledge Distillation.* Sequence-level knowledge distillation (Seq-KD) is a straightforward yet effective method that can achieve considerable performance improvements for NAT models. However, training an AT model as a teacher makes the training process redundant and limits the translation capabilities of

**Table 4: Performance of NAT4AT based on different AT and NAT models on WMT'14 EN↔DE benchmark.**

| Models | | Iter. | WMT'14 | |
|---|---|---|---|---|
| | | | EN→DE | DE→EN |
| MAT Model | GLAT | 1 | 25.97 | 30.01 |
| | GLAT w/ CandiSoups | 1 | 28.02 | 31.49 |
| | GLAT-Big | 1 | 27.03 | 30.81 |
| | GLAT-Big w/ CandiSoups | 1 | **28.86** | **31.99** |
| AT Model | Transformer-Base (beam = 1) | N | 28.69 | 32.20 |
| | Transformer-Big (beam = 1) | N | **29.03** | **32.60** |
| NAT4AT | GLAT + Transformer-Base | 6.1 | 28.88 | 32.56 |
| | GLAT w/ CandiSoups + Transformer-Base | 4.3 | 29.06 | 32.58 |
| | GLAT-Big + Transformer-Big | 5.6 | 29.32 | 32.84 |
| | GLAT-Big w/ CandiSoups + Transformer-Big | 4.1 | **29.43** | **32.91** |

**Table 5: Performance of NAT4AT without Seq-KD on WMT'16 EN↔RO benchmark.**

| Model | Iter. | WMT'16 | |
|---|---|---|---|
| | | EN→RO | RO→EN |
| AT (Transformer-Base) | N | 34.69 | 34.89 |
| GLAT + CandiSoups | 1 | 32.88 | 33.05 |
| NAT4AT | 4.8 | 34.95 | 35.14 |

NAT models. Therefore, many researchers have recently begun to focus on improving the performance of the NAT model in scenarios where Seq-KD is not used [13, 17]. To analyze whether our proposed method is still effective without using Seq-KD, we conduct experiments on the WMT'16 EN↔RO benchmark.

As shown in Table 5, when Seq-KD is not used, the performance of the NAT model drops significantly, about 2.0 BLEU lower than the AT model. However, NAT4AT can still generate better translations than the AT model with a small number of iterative decoding. This experimental result shows that even without Seq-KD, there is complementarity and high similarity between the translations generated by AT and NAT models, which makes our proposed NAT4AT method still very effective in this scenario. Due to space constraints, more analysis is presented in Appendix B.

## 6 RELATED WORK

With the emergence of Transformer [34] and various pre-trained models, such as MASS [32], T5 [21] and GPT3 [2], neural machine translation models achieve state-of-the-art performance on most machine translation benchmarks. However, they almost all predict translation in an autoregressive manner, resulting in high inference latency. To address the decoding inefficiency problem of autoregressive translation (AT), Gu et al. proposed non-autoregressive translation (NAT), which can decode the entire translation in parallel and achieve a superior inference speed. However, compared to the AT, its translation quality drops significantly.

Therefore, in order to improve the performance of the NAT, significant efforts have been made from various perspectives, including introducing latent variables [1, 30, 31, 39], training with better criterion [3, 13, 20, 25], using knowledge distillation [8, 29, 37] and iterative decoding [4, 5, 9]. Among them, the iterative NAT model will mask the wrong tokens in the output results of the previous iteration and then use them as the input in the next iteration to continuously refine and obtain the final translation. However, it is difficult to verify and revise the original translation only through the NAT model itself, so there is still an irreparable gap between the performance of the NAT model and the strong AT model.

There are also some methods [10, 23, 38] that try to combine AT and NAT to achieve better performance. For example, reorderNAT [35] uses the reordering information generated by the AT module to help the NAT decoding process, and Encoder-NAD-AD [38] uses the implicit global information generated by the NAT decoder to improve AT performance. These methods can improve translation quality but lead to a decrease in inference speed. In contrast, Xia et al. recently proposed a novel decoding method, generalized aggressive decoding (GAD), which can achieve lossless acceleration for the AT model with the help of an iterative NAT model (details are shown in Appendix A.1). Unlike them, our method takes advantage of the similarity and complementarity of AT and NAT translations, which can not only accelerate the inference speed but also improve the translation quality.

## 7 CONCLUSION

This paper finds complementarity and high similarity between AT and NAT translations through fine-grained analysis experiments. Based on this experimental result, we propose a general and effective method called NAT4AT, which uses the NAT model to generate original translation and uses the AT model to verify and revise it. Extensive experimental results show that our method can be applied to any AT and NAT model and significantly improves the translation quality and inference speed through the cooperation of the two. In addition, our best variant achieves excellent performance on two commonly used benchmarks while maintaining a **5.0×** speedup. In the future, we will further explore the performance of our method on other natural language generation tasks.

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

| Source Input | Von den neuen Einschränkungen sind junge Menschen , Minderheiten und Menschen mit niedrigem Einkommen unverhältnismäßig stark betroffen . |
|---|---|
| Input | [mask] [mask] [mask] [mask] [mask] [mask] [mask] [mask] [mask] [mask] |
| Draft NAT | The new restrictions **disproportionately** affect people , minorities and those |
| Verify AT | The new restrictions **inappropriately** affect young , minorities and those |
| Next Input | ==The new restrictions inappropriately== [mask] [mask] [mask] [mask] [mask] [mask] [mask] [mask] [mask] [mask] |

**Figure 7: An example shows how GAD works by using both NAT and AT models. Block size k = 10 in this example.**

**Table 6: Performance of ChatGPT and NAT4AT on WMT'14 EN↔DE test sets.**

| Model | Speedup | WMT'14 | |
|---|---|---|---|
| | | EN→DE | DE→EN |
| ChatGPT | 1.0× | 29.03 | **34.80** |
| NAT4AT | 43.6× | **29.06** | 32.58 |

## A  BACKGROUND

### A.1  Generalized Aggressive Decoding

Generalized Aggressive Decoding (GAD) was proposed by Xia et al., which decomposes an AT decoding iteration into two substeps: draft and verify, and each iteration can generate multiple tokens in parallel.

Figure 7 shows an example of how GAD works by using both NAT and AT models. Specifically, in the draft substep, GAD designed a draft NAT model to generate a block of tokens in parallel conditioning on the source sentence and previously verified tokens (==highlighted tokens==). In the verify substeps, GAD uses an AT model to verify drafted tokens and finds the first position that the drafted token does not match the top-1 result generated by AT and replaces it with AT's token. Then, all drafted tokens after this bifurcation position (red token) are discarded to ensure that GAD's decoding results will be exactly the same as AT. Then, in the next iteration, GAD continues to use the draft NAT model to generate a subsequent block of tokens in parallel and uses AT for verification.

Moreover, because top-1 matching is too strict, Xia et al. also proposed GAD++, which only requires drafted tokens to fall into top-k candidates with a tolerable score gap. We recommend that readers refer to the original paper for more details.

## B  ANALYSIS

### B.1  Influence of the Batch Size

We also test the inference speedup of NAT4AT for AT under different batch sizes. As shown in Figure 8, although the speedup of our method decreases with the increase of batch size, NAT4AT can still achieve a 3.3× speedup when batch size=32, which proves that our approach can still effectively accelerate the inference speed in the case of larger batch size.

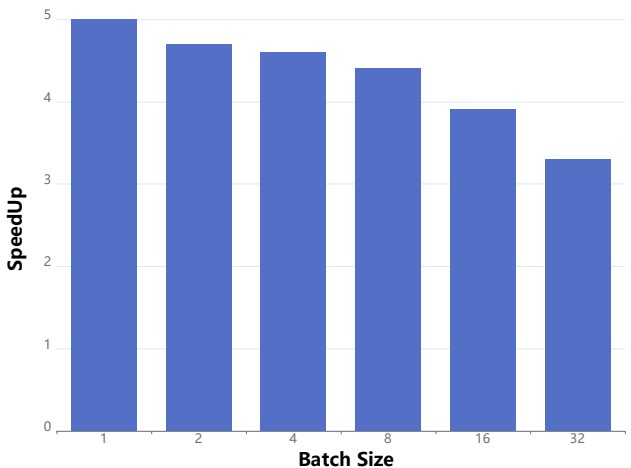

**Figure 8: The speedup of NAT4AT with different batch size on WMT'14 EN→DE test set. The speedup baseline is Transformer-Base (beam size=5).**

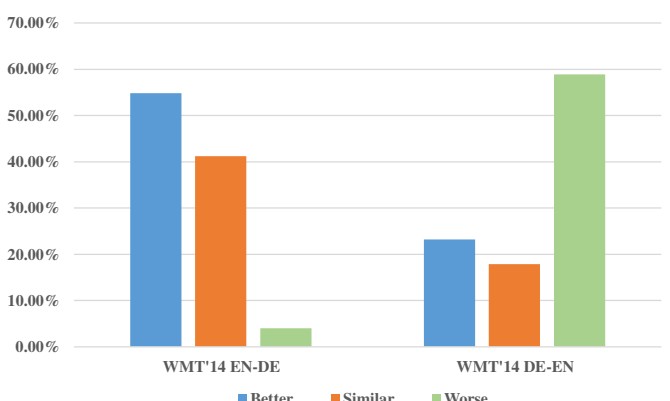

**Figure 9: Use ChatGPT to judge the quality of translations generated by NAT4AT and ChatGPT. Better indicates that ChatGPT believes that the translation generated by NAT4AT is better than ChatGPT itself.**

---

**Translation Prompt Template:**

Translate this source sentence from [source language] to [target language].

source sentence: [input]

target sentence:

---

**Evaluation Prompt Template:**

source sentence: [source]

reference translation: [reference]

candidate translation 1: [candidate1]

candidate translation 2: [candidate2]

Based on the given [source language] source sentence and [target language] reference translation above,

as well as the two [target language] candidate translations, determine which of the two candidate translations

is better. Returning 1 means candidate translation 1 is better, returning 2 means candidate translation 2 is better,

and returning 0 means both are of the similar quality.

Return:

---

**Figure 10: Prompt template for translation and evaluation of ChatGPT in our experiments.**

## B.2  Comparison with ChatGPT

Recently, large language models (LLMs) similar to ChatGPT[1] have achieved outstanding performance on machine translation tasks. Therefore, we compare the performance of our method with Chat-GPT on WMT'14 EN↔DE test sets. Our translation prompt template for ChatGPT can be found in Figure 10. The experimental results are shown in Table 6 and Figure 9.

As we can see in Table 6, NAT4AT achieves higher BLEU scores than ChatGPT on WMT'14 EN→DE translation task, while Chat-GPT performs better on DE→EN translation task. Considering that only the BLEU score may not be able to effectively measure the translation quality of ChatGPT, we also use ChatGPT itself for evaluation. Specifically, we wrote an evaluation prompt template (Figure 10), input the source sentence, reference translation, and two translations generated by ChatGPT and NAT4AT into ChatGPT, and let ChatGPT judge which of the two translations is better. As shown in Figure 9, NAT4AT can generate translations similar to or better than ChatGPT in most sentences on the EN→DE task. In contrast, on the DE→EN task, ChatGPT can generate better translations than NAT4AT on 58.9% sentences. We think this may be because ChatGPT's training corpus is almost English, which leads to its very strong ability to generate English translations. But in the EN→DE task, NAT4AT can not only generate better translations, but also the inference speed is 43.6 times that of ChatGPT, which further proves the effectiveness of our proposed method.

In addition, we conducted case studies on NAT4AT and ChatGPT. As shown in Figure 11, there is complementarity and high similarity between the translations generated by NAT and AT models. Therefore, NAT4AT can take advantage of these characteristics and generate high-quality translations similar to ChatGPT. Meanwhile, ChatGPT may generate hallucinated words that are never mentioned in the source sentence, but NAT4AT does not. We speculate that this is because the NAT model can only obtain information from the source sentence during inference, which may make its translation more faithful to the source sentence. The hallucination

problem is a challenge faced by existing LLMs and has attracted the attention of many researchers. And we will explore in the future whether our proposed approach can alleviate the hallucination problem in LLMs.

---

[1]https://chat.openai.com

**NAT4AT can generate high-quality translations similar to ChatGPT:**

| | |
|---|---|
| Source | Von den neuen Einschränkungen sind junge Menschen, Minderheiten und Menschen mit niedrigem Einkommen unverhältnismäßig stark betroffen. |
| ChatGPT | The new restrictions disproportionately affect young people, minorities, and individuals with low incomes. |
| NAT | The new restrictions disproportionately affect people, minorities and those those with low incomes. |
| AT | The new restrictions inappropriately affect young people, minorities and people with low income. |
| NAT4AT | The new restrictions disproportionately affect young people, minorities and those with low income. |

| | |
|---|---|
| Source | Arnold erklärte die Technik der neuen Anlage: Diese ist mit zwei Radarsensoren ausgestattet. |
| ChatGPT | Arnold explained the technology of the new system: it is equipped with two radar sensors. |
| NAT | Arnold explained the technique of the new plant: it is equipped with two radar sensors. |
| AT | Arnold explained the technology of the new plant: it is equipped with two radar sensors. |
| NAT4AT | Arnold explained the technology of the new plant: it is equipped with two radar sensors. |

**NAT4AT can generate translations that are more faithful to the source sentence:**

| | |
|---|---|
| Source | Diese Fahrer werden bald die Meilengebühren statt der Mineralölsteuer an den Bundesstaat zahlen. |
| ChatGPT | These drivers will soon pay mileage fees instead of the mineral oil tax to the state. |
| NAT | These drivers will soon pay the mileage fees instead of the oil tax to the state. |
| AT | These drivers will soon pay the mileage fees to the state instead of the oil tax. |
| NAT4AT | These drivers will soon pay the mileage fees instead of the oil tax to the state. |

| | |
|---|---|
| Source | Eine Entscheidung darüber wird voraussichtlich bereits im kommenden Jahr fallen. |
| ChatGPT | A final decision on this is expected to be made in the coming year. |
| NAT | A decision is expected to be made in the coming year. |
| AT | A decision on this is expected to be taken next year . |
| NAT4AT | A decision on this is expected to be made in the coming year. |

**Figure 11: Some cases show the translation results of ChatGPT and NAT4AT on WMT'14 DE→EN task.**

