# OpenReview forum: "NAT4AT: Using Non-Autoregressive Translation Makes Autoregressive Translation Faster and Better"
_ACM.org/TheWebConf/2024/Conference — TheWebConf24 Oral_

### Official Review · Reviewer_qSxC · 2023-11-17

**Novelty:** 5
**Technical Quality:** 5

**Review:**

This paper proposes a neural machine translation model, NAT4AT, which combines the advantages of both Autoregressive translation (AT) and Non-Autoregressive translation (NAT) models to speed up translation without losing translation quality. In this paper, the feasibility of combining AT and NAT to improve the translation speed and quality is analyzed through experiments, and then the idea and detailed process of NAT4AT method are described. Finally, the advantages of NAT4AT method are confirmed through the comparison experiments with related work and experiments with different parameter settings.
Overall, this paper is well expressed and readable, and the experimental design is comprehensive. In terms of details, there are some areas that need further improvement.
1. The references are not standardized enough. For example, on page 2, left column lines 126-127, “Xia et al. proposed generalized aggressive decoding (GAD)” should give a reference. Even though it was given in the experimental part, this is the first time it's been shown. Similar problems exist elsewhere in the paper.
2. To ensure reproducibility of the experiment, the datasets used in the experiment should give references or URL.
3. The sliding window algorithm is mentioned in the right column of page 4, line 447, which should be marked in Algorithm 1, or its corresponding position in Algorithm 1 should be pointed out in the text of paper.
In addition, I am not an expert in machine translation, so I have some questions for you. See Question.

**Questions:**

1. According to your paper, AT translation quality is good but slow, NAT translation speed is fast but not good, so NAT4AT combines them. However, in Introduction and Related Work, you did not introduce the model related to AT, and the Transformer-Base and Transformer-Big mentioned in the experiment did not give references. My question is, why are the experimental results in Table 2 not compared with other pure AT models?
2. In Table 2 and 6, Speedup metric is based on Transformer-Base model, so why not just give the absolute speed?
3. Other than BLEU, are there any other metrics for evaluating translation quality?
4. For machine translation, LLMs are a topic that cannot be sidekicked. You paper also compares NAT4AT with ChatGPT, and NAT4AT has some advantages. So should LLMs be mentioned in Experiment to guide readers to Appendix B? In the meantime, I suggest that the results of Table 6 be added to Table 2, and LLMs should mention in Conclusion.

**Reviewer Confidence:**

2: The reviewer is willing to defend the evaluation, but it is likely that the reviewer did not understand parts of the paper

**Scope:**

4: The work is relevant to the Web and to the track, and is of broad interest to the community

---

### Official Review · Reviewer_z1gq · 2023-11-20

**Novelty:** 5
**Technical Quality:** 5

**Review:**

This paper proposes simple heuristics to use autoregressive translation models (AT) to alter the output of non-autoregressive translation models (NAT), which leads to better and faster translations. The authors conducted preliminary experiments to show that existing AT and NAT methods usually have overlapping words in translations. Based on this observation, they propose to use AT to correct the output of NAT to get the best of both worlds (i.e., the speed of NAT and the accuracy of AT).

## Pros

- The paper is well written and easy to follow.
- The proposed method is sound, well motivated, and effective.
- The experiments are thorough.

## Cons

- The method is quite simple and as I understand, only works for sentences. It would be interesting to see how to extend it to longer texts, I guess the trick in lines 382-383 would not work as well as for sentences.

## After Rebuttal

I have read the author responses and would like to keep my evaluation scores.

**Questions:**

I get the "faster" part. However, it's not quite intuitive to me why the proposed method translates "better" than AT, because the method only replaces words in the NAT translations if the AT model provides words with higher probabilities. Hence, the AT translation quality might be the upper bound here. Please clarify if I misunderstood something.

**Reviewer Confidence:**

3: The reviewer is confident but not certain that the evaluation is correct

**Scope:**

4: The work is relevant to the Web and to the track, and is of broad interest to the community

---

### Official Review · Reviewer_3Br9 · 2023-11-23

**Novelty:** 6
**Technical Quality:** 6

**Review:**

Pros:
-  The paper conducts a detailed analysis at the sentence level. The authors address fine-grained questions about the similarities between AT and NAT translations
- The results demonstrate the generality and effectiveness of NAT4AT on major WMT benchmarks


Cons:
- The paper does not have a strong theoretical foundation
- A more rooted sensitivity analysis for the threshold's impact would have been useful

**Questions:**

What are the major contributing factors for the speedup? A  deeper complexity analysis would have been helpful?

**Reviewer Confidence:**

4: The reviewer is certain that the evaluation is correct and very familiar with the relevant literature

**Scope:**

4: The work is relevant to the Web and to the track, and is of broad interest to the community

---

### Official Review · Reviewer_Lzoy · 2023-11-23

**Novelty:** 5
**Technical Quality:** 6

**Review:**

This paper proposes a method for accelerating the inference speed of an auto-regressive translation(AT) model using another non-auto-regressive translation(NAT) model, while retaining or even improving AT model's translation quality.

Pros:
1. The preliminary experiments on the sentence-level translation quality and LCS-bases similarity reveals important characteristics of NAT and AT models. The observation then motivate the proposed NAT4AT method for exploiting the complementarity between NAT and AT.
2. The experimental results confirm the advantage of the proposed NAT4AT in terms of quality-efficiency tradeoff, compared to fully NAT, fully AT, and iterative NAT methods.
3. The ablations are comprehensive and useful.

Cons:
1. More experiments upon multilingual Large Language Models(LLMs) are missing.
2. Discussion comparing NAT4AT with Speculative Decoding is missing. There two are highly relevant since they both hinge on a more efficient model in a draft-then-verify manner.

**Questions:**

1. What is the additional memory overhead incurred by NAT4AT?
2. Any comparison with Speculative decoding? For example, using a small AT model as the draft model and a larger AT model as the target model.

**Reviewer Confidence:**

3: The reviewer is confident but not certain that the evaluation is correct

**Scope:**

4: The work is relevant to the Web and to the track, and is of broad interest to the community

---

### Official Review · Reviewer_ddPN · 2023-11-24

**Novelty:** 6
**Technical Quality:** 5

**Review:**

### Summary

This paper proposes a method called NAT4AT, which uses a non-autoregressive translation (NAT) model to generate an initial translation and an autoregressive translation (AT) model to correct errors in the initial translation. The paper claims that NAT4AT can achieve better translation quality and faster inference speed than existing NAT and AT methods. The paper also conducts analysis experiments to show the similarity and complementarity between AT and NAT translations at the sentence level.

### Strengths

The paper addresses an important problem of improving the efficiency and effectiveness of neural machine translation. The paper presents a novel and simple idea of using NAT to assist AT, which is orthogonal to the model architecture and can be applied to any NAT and AT models. The paper provides extensive experimental results on two translation benchmarks, demonstrating the superiority of the proposed method over strong baselines. The paper also provides detailed analysis and ablation studies to verify the impact of different components and hyperparameters of the method.

### Weaknesses

The paper lacks some analysis and justification for the proposed method. For example, why does the threshold s and the window width $w$ affect the performance of NAT4AT? How to choose the optimal values for these hyperparameters? How does NAT4AT deal with the length prediction and word alignment issues?

**Questions:**

See weaknesses.

**Reviewer Confidence:**

3: The reviewer is confident but not certain that the evaluation is correct

**Scope:**

3: The work is somewhat relevant to the Web and to the track, and is of narrow interest to a sub-community

---

### Decision · Program_Chairs · 2024-01-22

**Decision:**

Accept (Oral)

**Comment:**

The paper presents a novel and simple idea of using non-autoregressive translation (NAT) to generate an initial translation and then using autoregressive translation (AT) to correct errors in the initial translation. This can achieve better translation quality and faster inference speed than existing NAT and AT methods. The paper conducts analysis experiments to show the similarity and complementarity between AT and NAT translations at the sentence level.

 The paper addresses an important problem of improving the efficiency and effectiveness of neural machine translation, and that the proposed method is sound, well motivated, and effective. There are extensive experimental results on two translation benchmarks, demonstrating the superiority of the proposed method over strong baselines, as well as detailed analysis and ablation studies to verify the impact of different components and hyperparameters of the method.

 There are a few points raised by the reviewers, many of which have been addressed by the authors. I recommend the authors modify the draft to include these additional results and explanations.

 Given the reviews and the author responses, I recommend an accept. I think the paper is borderline, but has enough going for it to be part of the WebConf proceedings.